# Improving Segmentation of Objects with Varying Sizes in Biomedical Images using Instance-wise and Center-of-Instance Segmentation Loss Function

**Muhammad Febrian Rachmadi**[1,2]                 FEBRIAN.RACHMADI@RIKEN.JP

**Charissa Poon**[1]                 CHARISSA.POON@RIKEN.JP

**Henrik Skibbe**[1]                 HENRIK.SKIBBE@RIKEN.JP

[1] *Brain Image Analysis Unit, RIKEN Center for Brain Science, Wako, Japan*

[2] *Faculty of Computer Science, Universitas Indonesia, Depok, Indonesia*

**Editors:** Accepted for publication at MIDL 2023

## Abstract

In this paper, we propose a novel two-component loss for biomedical image segmentation tasks called the Instance-wise and Center-of-Instance (ICI) loss, a loss function that addresses the instance imbalance problem commonly encountered when using pixel-wise loss functions such as the Dice loss. The Instance-wise component improves the detection of small instances or "blobs" in image datasets with both large and small instances. The Center-of-Instance component improves the overall detection accuracy. We compared the ICI loss with two existing losses, the Dice loss and the blob loss, in the task of stroke lesion segmentation using the ATLAS R2.0 challenge dataset from MICCAI 2022. Compared to the other losses, the ICI loss provided a better balanced segmentation, and significantly outperformed the Dice loss with an improvement of $1.7 - 3.7\%$ and the blob loss by $0.6 - 5.0\%$ in terms of the Dice similarity coefficient on both validation and test set, suggesting that the ICI loss is a potential solution to the instance imbalance problem.

**Keywords:** Instance-wise and Center-of-Instance segmentation loss, segmentation loss.

## 1. Introduction

Object segmentation in biomedical images is a common task, yet presents challenges, namely class imbalance and instance imbalance problems, due to the diversity of object sizes. Class imbalance problem happens when the number of pixels of a class is much higher than the other classes. Whereas, instance imbalance problem happens when larger instances dominates over smaller instances of the same class. These are two frequent problems that arise when objects appear as multiple instances of diverse sizes in an image, such as stroke lesions in brain magnetic resonance imaging (MRI) (Guerrero et al., 2018). Addressing these challenges is crucial for accurate segmentation and improved diagnostic results.

Deep learning semantic segmentation models often utilize a pixel-wise loss function to evaluate the quality of the segmentations produced by the model across the whole image. Previous studies have demonstrated that pixel-wise loss functions such as cross-entropy (CE) and Dice losses are effective at segmenting large objects and instances (Ronneberger et al., 2015; Milletari et al., 2016). However, pixel-wise loss functions have difficulty in identifying instances of varying sizes, as they tend to miss small features within large instances, and fail to detect individual small instances (Jeong et al., 2019; Maulana et al., 2021). This

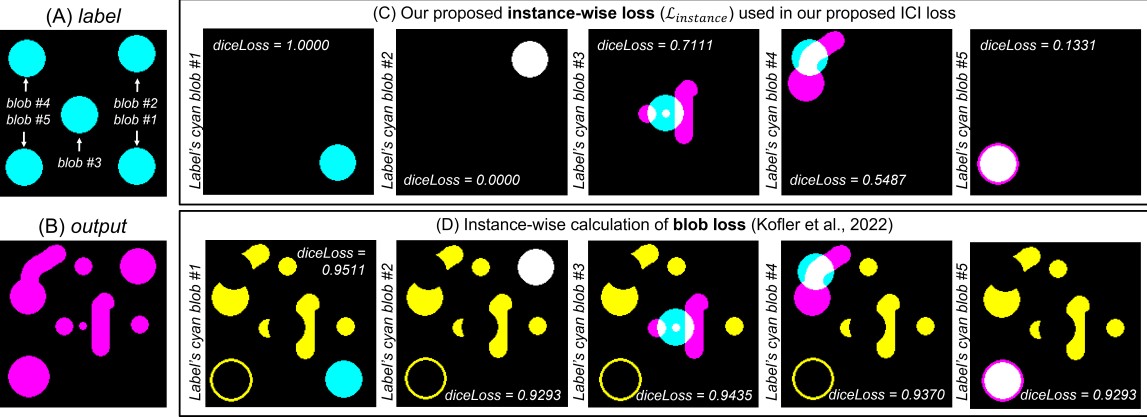

Figure 1: Comparison of instance-wise segmentation losses performed by our proposed Instance-wise loss ($\mathcal{L}_{instance}$) (C) and the blob loss (Kofler et al., 2022) (D). Artificial data are used for clearer visualization, where (A) shows the *label* image, consisting of 5 instances (cyan), and (B) shows the *output* image with many more instances (magenta). Each column in (C) and (D) represents an individual instance from the *label* image. In (C), our proposed Instance-wise loss only includes false segmentations from *output* instances that have intersections with the instance-of-interest (magenta). In contrast, blob loss (D) includes false segmentations from other *output* instances (yellow). White are correct segmentations.

is mainly due to their focus on individual pixels, rather than considering the context of objects in an image (Reinke et al., 2021).

Recent advancements in biomedical image analysis have led to the development of new loss functions (Ma et al., 2021), e.g. Tversky loss (Salehi et al., 2017), Focal loss (Lin et al., 2017), Generalized Dice loss (Sudre et al., 2017), and a combination of Dice loss with CE loss (Isensee et al., 2021), all designed to address the class imbalance problem. However, most of these methods are pixel-wise and fail to tackle the instance imbalance problem.

The instance imbalance problem, where larger instances dominate smaller instances, remains a significant challenge. A normalized instance-wise loss function, where a loss value is computed for each instance individually, is required to address this problem. In this study, **our main contribution is the proposal of a novel instance-wise compound loss function, named the Instance-wise and Center-of-Instance (ICI) loss function**, which can be utilized in conjunction with any pixel-wise loss function for regularization. Through experimental evaluation, we demonstrate the superior performance of the ICI loss function compared to other related loss functions.

## 2. Related Approaches

Several loss functions have been proposed to solve the instance imbalance problem in biomedical image segmentation, including inverse weighting (Shirokikh et al., 2020), the blob loss (Kofler et al., 2022), and the lesion-wise loss (Zhang et al., 2021).

**Inverse weighting (IW)** was proposed as an instance-weighted loss function, where each instance is given a weight that is inversely proportional to its size by using a global weight map (Shirokikh et al., 2020). However, IW is implemented by assigning weights to each individual pixel, and is not computed on each instance separately.

In contrast, the **blob loss** is an instance-wise loss calculated for each instance in the label and averaged over all instances (Kofler et al., 2022). But the blob loss is overly sensitive to false segmentations as it includes false segmentations from other instances, even if they do not intersect with the instance-of-interest (see Figure 1 for visualization). This is because connected component analysis (CCA) is precomputed outside of the blob loss and only for the label image, so the blob loss cannot distinguish which instances of the output image (predicted segmentation) intersect with each instance of the label image (ground truth).

Lastly, the **Lesion-wise loss (LesLoss)** was proposed to assign each blob the same size by transforming all instances into spheres with a fixed size based on instances' centers of mass (Zhang et al., 2021). Similar to the blob loss, the CCA is precomputed outside of the LesLoss and only for the label image. Also, an additional segmentation network is needed to perform segmentation of instances' spheres which limits its applicability.

## 3. Proposed Approach

Our proposed **Instance-wise and Center-of-Instance (ICI) loss function** is inspired by both the blob loss (Kofler et al., 2022) and LesLoss (Zhang et al., 2021), where we combine instance-wise loss calculation with the normalization of all instances into a square/cube (2D/3D images) of a fixed size. The ICI loss function consists of two loss calculations: the **Instance-wise loss** and the **Center-of-Instance loss** calculations. In general, the Instance-wise loss, distinct from the existing blob loss (Kofler et al., 2022), is used to minimize missed segmentations of instances in the label image (reduce false negatives), especially the small instances. Whereas, the Center-of-Instance loss is used to improve the segmentation of small instances in the label image (reduce false negatives) and suppress the detection of small and spurious instances in the output image (reduce false positives).

In this study, we used a compound loss which combines the global Dice loss ($\mathcal{L}_{global}$), Instance-wise loss ($\mathcal{L}_{instance}$), and Center-of-Instance loss ($\mathcal{L}_{center}$) with weights $a$, $b$, and $c$, respectively. The compound loss is shown in Equation (1). The Dice loss is also used to calculate the individual components of the ICI loss; the Dice loss equation is shown in Equation (2). The general flow chart and visualization of the ICI loss is shown in Figure 2 and the formalism of our proposed ICI loss can be seen in Appendix A.

$$\mathcal{L} = a \times \mathcal{L}_{global} + b \times \mathcal{L}_{instance} + c \times \mathcal{L}_{center} \tag{1}$$

$$diceLoss = 1 - \frac{2|y \cap y_{\text{pred}}|}{|y| + |y_{\text{pred}}|} = 1 - \frac{2\,TP}{2\,TP + FP + FN} \tag{2}$$

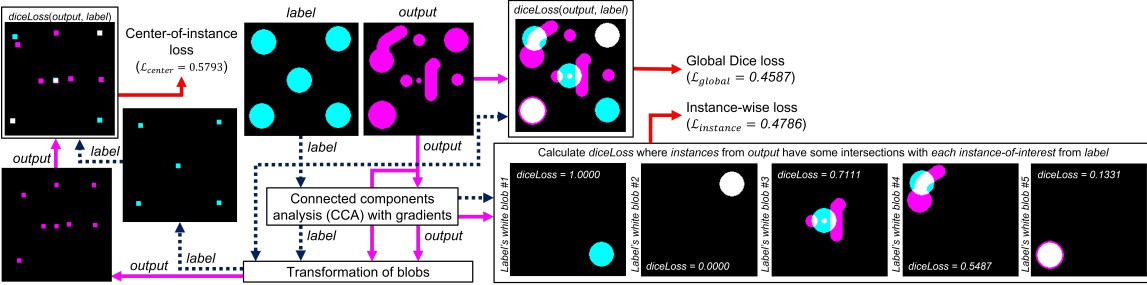

Figure 2: Flow chart and visualization of all losses used in this study. The total loss ($\mathcal{L}$) used in this study is a compound loss, as formulated in Equation (1), consisting of the global Dice loss ($\mathcal{L}_{global}$) and our proposed ICI loss which are Instance-wise loss ($\mathcal{L}_{instance}$) and Center-of-Instance loss ($\mathcal{L}_{center}$).

### 3.1. Instance-wise loss

Our proposed **Instance-wise loss ($\mathcal{L}_{instance}$)** calculation is similar to the blob loss, where an instance-wise segmentation loss is performed by computing the Dice loss for each instance in the manual label. The difference between our Instance-wise loss calculation and the blob loss is that CCA is computed on the fly in our proposed loss function (on GPUs) for both the manual labels and the predicted segmentations. By performing CCA on the fly for both labels and predictions, all instances in the predicted segmentation that overlap by at least 1 pixel/voxel with manually labelled instances can be identified. This approach is not performed in the blob loss, which leads to the inclusion of false segmentations that do not have any overlap with the instance (in the manual label) being calculated (see Figure 1).

Instance-wise loss calculation consists of two steps: 1) performing CCA for both the manual label (*label*) and predicted segmentation produced by a deep learning model (*output*), and 2) calculating the Dice loss for each instance in the *label* (instance-of-interest) against any instances in the *output* that intersect with the instance-of-interest. We implemented CCA by modifying the *connected_components* function from the kornia library (Riba et al., 2020) so that all gradients in the predicted segmentation are tracked for backpropagation. In this study, the threshold value of 0.5 was heuristically determined and used to threshold the predicted segmentations before performing CCA. After all instances in *label* (i.e. *cc_label*) and *output* (i.e. *cc_output*) are identified, the Instance-wise loss can be computed by following Algorithm 1 in Appendix E.

### 3.2. Center-of-Instance loss

Our proposed **Center-of-Instance loss ($\mathcal{L}_{center}$)** calculation transforms all blobs into squares (2D) or cubes (3D) with a fixed size before the Dice loss is calculated. Squares/cubes are used instead of circles/spheres (used in the LesLoss) to simplify the transformation's calculation. Simple calculation is important because all gradients need to be tracked for backpropagation on GPUs. All instances in *label* and *output* images can be easily transformed into squares/cubes as the center-of-mass for each instance is calculated during CCA.

Using the center-of-mass, each instance can then be transformed into a square/cube by assigning 1 to the pixels/voxels that make up the square/cube area. Visualization of the transformation of 2D blobs into 2D squares ($7 \times 7$ pixels) is shown in Figure 2. Visualization in 3D space can be found in Appendix H. Based on our preliminary experiments, the best size of the fixed-size cubes for an image of original size $192 \times 192 \times 192$ voxels was found to be $7 \times 7 \times 7$ voxels (see Table 5 in Appendix I). Pseudo-code of the Center-of-Instance loss calculation is shown in Algorithm 2 in Appendix F.

## 4. Experimental Settings

All experimental settings used in this study are described below.

**Deep learning model:** For all experiments, we used a 3D Residual U-Net (Kerfoot et al., 2018) loaded from the MONAI library. Parameters that were used to create the 3D Residual U-Net are shown in Appendix D. We used a sigmoid function with 0.5 as the threshold value in the segmentation layer for binary segmentation.

**Tested loss functions:** We compared our proposed ICI loss directly to the blob loss (Kofler et al., 2022) and the global Dice loss. We did not perform any comparisons with LesLoss because an additional network is needed for predicting each instance's sphere. For the blob loss, we used the recommended weights, which are $\alpha = 2$ for the (main) global Dice segmentation loss and $\beta = 1$ for the instance-wise segmentation, which also uses the Dice loss. For our proposed compound ICI loss, we tested different weights for the global Dice loss ($a$), the Instance-wise loss ($b$), and the Center-of-Instance loss ($c$).

**Dataset:** We used the publicly available ATLAS v2.0 challenge dataset from the MIC-CAI 2022 challenge (Liew et al., 2022) available at https://atlas.grand-challenge.org/. ATLAS v2.0 is a large public dataset of T1w stroke brain MRI and manually segmented lesion masks ($N = 1,271$) that is divided into a public training set ($n = 655$), a test set where the lesion masks are hidden ($n = 300$), and a generalizability set, where both T1w and lesions masks are hidden ($n = 316$). Specific to this study, we only used the public training set for training and validation, and the test set for testing the trained models. We did not use the generalizability set because only one model can be submitted to the system per month. In contrast, one submission can be submitted per day for evaluating the test set.

**Training:** Out of 655 subjects in the public training set from the ATLAS v2.0 challenge dataset, we manually divided it into a training set ($n = 600$) and validation set ($n = 55$). We performed two different experiments: whole image experiments (with image size of $192 \times 192 \times 192$), and patch-based experiments (with patch size of $96 \times 96 \times 96$) to assess the effectiveness of our proposed ICI loss in segmenting 3D images of different sizes. In the whole image experiment, we trained 3D Residual U-Net models for 200 epochs by using a mini-batch of 4 random subjects in every step. In the patch-based experiment, we trained 3D Residual U-Net models for 600 epochs (with mini-batches of 2 subjects), where 8 patches were randomly extracted from each subject, with a 1:1 ratio for positive (stroke lesions) and negative (non-stroke lesions) labels. Randomized data augmentations were applied, including left/right flipping, rotation, zooming, and intensity scaling and shifting. Subject-wise intensity normalization was performed by using zero mean unit variance. The 3D Residual U-Net was optimized using the Adam optimizer (Kingma and Ba, 2015). The model that produced the best Dice metric in the validation set was used in testing.

**Training environments:** We conducted our experiments using various NVIDIA GPUs, including rtxa6000, rtxa5000, v100, a100, and rtx8000, with CUDA version 11.7, Pytorch version 1.13.0, and MONAI version 0.9.0.

**Testing/inference:** We first performed inference on T1w brain MRI in the test set on our computing nodes, and then we submitted the predicted segmentation results to the ATLAS v2.0 challenge's system. Note that only one submission was permitted per day.

**Performance measurements:** The ATLAS v2.0 challenge produced 4 performance measurements: Dice similarity coefficient (DSC), volume difference, lesion-wise F1 score, and simple lesion count. We also measured the performance of all models in the validation set by using our own performance measurements, which are DSC, total and numbers of subjects with missed instances (MI), total and numbers of subjects with false instances (FI), and subjects without MI & FI. To decide which model performed best, a numeric rank (written inside square brackets [ ]) is given to each performance measurement such that a mean rank for each model can be calculated.

## 5. Results

In this section, the ↑ symbol means that higher values are better, while the ↓ symbol means that lower values are better. The best value for each column is shown in bold and the second best is underlined.

### 5.1. Whole image segmentation

Figure 3 shows that compounding a pixel-wise segmentation loss (i.e., Dice loss) with both terms of our proposed ICI loss (i.e., in $a = 1, b = 1, c = 1$ (yellow lines) and $a = 1/4, b = 1/2, c = 1/4$ (purple lines)) have several advantages in training and validation compared to the other losses. First, yellow and purple lines achieved lower Dice losses in fewer training epochs, as shown in Figure 3A. Second, yellow and purple lines produced lower and more stable numbers of missed and false instances than the other losses during validation, as shown in Figure 3B and C. Lastly, yellow and purple lines achieved higher DSC values more quickly than the other losses during validation as shown in Figure 3D. All these observations suggest that ICI loss successfully regularized Dice loss by keeping the number of missed and false instances low, in addition to lowering the Dice loss itself.

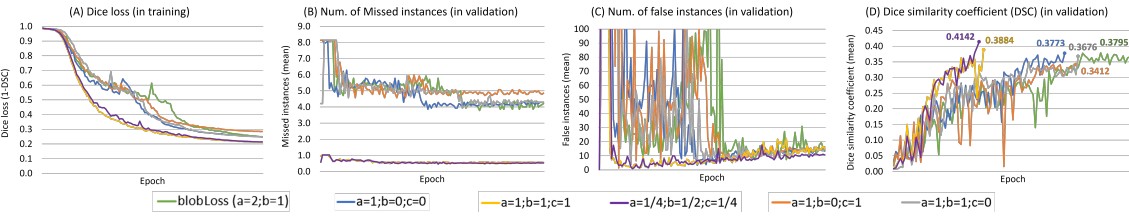

Figure 3: Training curves for (A) Dice loss and validation curves for (B) number of missed blobs, (C) number of false blobs, and (D) DSC values.

Table 1: Performance values on the validation set from the whole image experiments.

| Weights (a=global, b=blob, c=center) | Mean Rank ($\downarrow$) | DSC ($\uparrow$) | Total MI ($\downarrow$) | Subjects w/ MI ($\downarrow$) | Subjects w/ all MI ($\downarrow$) | Total FI ($\downarrow$) | Subjects w/ FI ($\downarrow$) | Subjects wo/ MI & FI ($\uparrow$) | Best Epoch |
|---|---|---|---|---|---|---|---|---|---|
| blob loss ($\alpha = 2, \beta = 1$) | 4.00 | 0.3795 [3] | 53 [2] | 29 [1] | 11 [1] | 382 [6] | 53 [5] | 0 [6] | 107 |
| a = 1, b = 0, c = 0 | 3.67 | 0.3773 [4] | 58 [5] | 32 [3] | 14 [3] | 149 [3] | 46 [2] | 4 [2] | 77 |
| a = 1, b = 0, c = 1 | 3.83 | 0.3412 [6] | 67 [6] | 36 [4] | 19 [4] | 110 [1] | 37 [1] | 5 [1] | 82 |
| a = 1, b = 1, c = 0 | 3.67 | 0.3676 [5] | 54 [3] | 29 [1] | 12 [2] | 205 [4] | 51 [4] | 3 [3] | 83 |
| a = 1, b = 1, c = 1 | 3.50 | 0.3884 [2] | 51 [1] | 30 [2] | 11 [1] | 225 [5] | 54 [6] | 1 [4] | 41 |
| a = 1/4, b = 1/2, c = 1/4 | 2.83 | 0.4142 [1] | 55 [4] | 32 [3] | 11 [1] | 124 [2] | 49 [3] | 3 [3] | 39 |

Table 2: Performance values on the test set from the whole image experiments.

| Weights (a=global, b=blob, c=center) | Mean Rank ($\downarrow$) | DSC ($\uparrow$) | Volume Difference ($\downarrow$) | Lesion-wise F1 Score ($\uparrow$) | Simple Lesion Count ($\downarrow$) |
|---|---|---|---|---|---|
| blob loss ($\alpha = 2, \beta = 1$) | 5.50 | 0.3954 (0.3058) [5] | 18,358.73 (34,566.44) [6] | 0.3036 (0.2374) [5] | 6.7900 (29.7589) [6] |
| $a = 1, b = 0, c = 0$ | 2.50 | 0.4147 (0.3097) [4] | 16,437.22 (28,680.58) [3] | 0.3558 (0.2638) [2] | 4.5367 (6.1965) [1] |
| $a = 1, b = 0, c = 1$ | 4.25 | 0.3904 (0.3209) [6] | 16,148.17 (29,651.68) [2] | 0.3028 (0.2579) [6] | 4.9967 (6.7012) [5] |
| $a = 1, b = 1, c = 0$ | 3.75 | 0.4160 (0.2991) [3] | 17,782.27 (34,580.36) [5] | 0.3233 (0.2334) [4] | 4.7367 (6.1601) [3] |
| $a = 1, b = 1, c = 1$ | 3.25 | 0.4240 (0.3067) [2] | 16,717.15 (27,592.10) [4] | 0.3508 (0.2373) [3] | 4.9667 (6.4584) [4] |
| $a = 1/4, b = 1/2, c = 1/4$ | 1.25 | 0.4455 (0.3099) [1] | 15,993.67 (30,064.58) [1] | 0.4147 (0.2731) [1] | 4.6900 (6.8963) [2] |

Quantitative results for all losses in the validation set produced by using the best models (i.e., with the highest DSC values) can be seen in Table 1. Note that $\mathcal{L}_{center}$ itself ($a = 1, b = 0, c = 1$) produced the lowest total FI, but failed to produced lower total MI and higher DSC. On the other hand, compounding Dice loss with all of ICI loss's terms with optimum weights ($a = 1/4, b = 1/2, c = 1/4$) ranked the best by producing the best DSC with mean rank of 2.83. In contrast, blob loss with recommended weights ($\alpha = 2, \beta = 1$) failed to produce better results except for the number of subjects with MI in the validation set, showing a mean rank of 4.00, which is worse than the baseline Dice loss without regularization ($a = 1, b = 0, c = 0$) which showed a mean rank of 3.67.

Table 2 shows that compounding Dice loss with all ICI loss's terms with optimum weights ($a = 1/4, b = 1/2, c = 1/4$) is quite robust to the unseen test set by producing the best values for DSC, Volume Difference, and Lesion-wise F1 Score, and the second best value for Simple Lesion Count with the highest rank (mean rank of 1.25). In contrast, blob loss with recommended weights ($\alpha = 2, \beta = 1$) failed again to achieve a better mean rank than the baseline Dice loss without regularization ($a = 1, b = 0, c = 0$).

## 5.2. Patch-based segmentation

Our proposed ICI loss also showed superior performance in the patch-based segmentation task when compared to both the baseline Dice loss without regularization ($a = 1, b = 0, c = 0$) and the blob loss with recommended weights ($\alpha = 2, \beta = 1$) in both the validation and test sets, as seen in Tables 3 and 4. However, note that the optimum weights of the ICI loss used in the whole image segmentation experiments (i.e., $a = 1/4, b = 1/2, c = 1/4$) were not the best in terms of mean rank in the test set, suggesting that the optimal weights for the ICI loss may be dependent on the input image size and task. Nevertheless, Table 3 and Table 4 show that the ICI loss is robust to the unseen test set, as different weights of the ICI loss consistently achieved higher mean ranks compared to the baseline Dice loss without regularization in both validation and test sets.

Table 3: Performance values on the validation set from the patch-based experiments.

| Weights (a=global, b=blob, c=center) | Mean Rank ($\downarrow$) | DSC ($\uparrow$) | Total MI ($\downarrow$) | Subjects w/ MI ($\downarrow$) | all MI ($\downarrow$) | Total FI ($\downarrow$) | Subjects w/ FI ($\downarrow$) | Subjects wo/ MI & FI ($\uparrow$) | Best Epoch |
|---|---|---|---|---|---|---|---|---|---|
| blob loss ($\alpha = 2, \beta = 1$) | 3.67 | 0.5237 [3] | 34 [4] | 24 [3] | 6 [2] | 471 [3] | 53 [4] | 1 [3] | 403 |
| a = 1, b = 0, c = 0 | 4.33 | 0.5124 [5] | 30 [2] | 24 [3] | 7 [3] | 503 [5] | 54 [5] | 1 [3] | 292 |
| a = 1, b = 0, c = 1 | 3.83 | 0.5082 [6] | 32 [3] | 24 [3] | 6 [2] | 478 [4] | 51 [2] | 1 [3] | 387 |
| a = 1, b = 1, c = 0 | 3.17 | **0.5310** [1] | 36 [5] | 27 [4] | 6 [2] | 462 [2] | 52 [3] | 2 [2] | 440 |
| a = 1, b = 1, c = 1 | 2.17 | 0.5201 [4] | 30 [2] | 23 [2] | 6 [2] | **441** [1] | **50** [1] | **3** [1] | 422 |
| a = 1/4, b = 1/2, c = 1/4 | **2.33** | 0.5295 [2] | **27** [1] | **22** [1] | **5** [1] | 513 [6] | 51 [2] | **3** [1] | 422 |

Table 4: Performance values on the test set from the patch-based experiments.

| Weights (a=global, b=blob, c=center) | Mean Rank ($\downarrow$) | DSC ($\uparrow$) | Volume Difference ($\downarrow$) | Lesion-wise F1 Score ($\uparrow$) | Simple Lesion Count ($\downarrow$) |
|---|---|---|---|---|---|
| blob loss ($\alpha = 2, \beta = 1$) | 5.25 | 0.5754 (0.2743) [4] | 12,210.94 (24,185.93) [5] | 0.4033 (0.2413) [6] | 5.9933 (7.9349) [6] |
| a = 1, b = 0, c = 0 | 5.00 | 0.5598 (0.2724) [6] | 13,857.66 (28,911.29) [6] | 0.4092 (0.2497) [5] | 4.9900 (6.2128) [3] |
| a = 1, b = 0, c = 1 | 4.00 | 0.5713 (0.2764) [5] | 12,031.63 (24,436.64) [4] | 0.4263 (0.2633) [3] | 5.3000 (6.6907) [4] |
| a = 1, b = 1, c = 0 | 2.50 | 0.5805 (0.2822) [2] | **10,978.48 (22,846.95)** [1] | 0.4314 (0.2621) [2] | 5.3800 (6.7756) [5] |
| a = 1, b = 1, c = 1 | **1.75** | 0.5790 (0.2717) [3] | 11,564.10 (23,959.23) [2] | **0.4480 (0.2557)** [1] | **4.7800 (6.5443)** [1] |
| a = 1/4, b = 1/2, c = 1/4 | 2.50 | **0.5817 (0.2705)** [1] | 11,588.88 (23,726.86) [3] | 0.4256 (0.2362) [4] | 4.9867 (6.1807) [2] |

## 6. Conclusion

This paper presents a novel Instance-wise and Center-of-Instance (ICI) loss which improved the segmentation of multiple instances with various sizes in biomedical images. In this study, we compared our ICI loss with the Dice loss, a popular pixel-wise segmentation loss, and the blob loss, which was proposed as an instance-wise segmentation loss, in the task of stroke lesion segmentation on the ATLAS R2.0 challenge dataset from MICCAI 2022. Our experiments show that using the ICI loss led to an average increase of 2.7% in segmentation accuracy compared to the Dice loss and 2.4% compared to the blob loss in both whole image segmentation and patch-based segmentation. The codes (implementation) of ICI loss in Pytorch is available at (https://github.com/BrainImageAnalysis/ICI-loss).

There are many applications in biomedical image analysis in which the ICI loss may be useful, because many objects that are common targets of segmentation tasks consist of multiple instances of various sizes. The ICI loss has similar limitations to the blob loss: specifically, additional computational resources are required for performing CCA, and performance of the loss function may be sensitive to the weights and hyperparameters used. In our experiments with batch of $4 \times 1 \times 192 \times 192 \times 192$, Dice loss, blob loss, and our proposed ICI loss took 0.26, 1.24, and 2.19 seconds to finish all computations per batch, respectively (see Appendix G for further analysis). Furthermore, our experiments have shown that a simple set of weights $a = 1, b = 1, c = 1$, without extensive hyperparameter tuning, is sufficient to improve segmentation results in all of the cases based on the DSC metric. The next step is to evaluate whether the ICI loss performs well in multi-class segmentation problems where some classes present as multiple instances with various sizes while others do not. Furthermore, combination with other pixel-wise losses such as CE and Boundary losses might be explored in future studies (some preliminary results can be observed in Appendix J).

## Acknowledgments

This work was supported by the program for Brain Mapping by Integrated Neurotechnologies for Disease Studies (Brain/MINDS) from the Japan Agency for Medical Research and Development AMED (JP15dm0207001). Library access provided by the Faculty of Computer Science, Universitas Indonesia is also gratefully acknowledged. CP was also supported by the Grant-in-Aid for Scientific Research for Young Scientists (KAKENHI 22K15658).

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

## Appendix A. Formalism of Instance-wise and Center-of-Instance losses

Let $\Omega$ be the image domain, and let $y_c : \Omega \to \{0, 1\}$ be a binary mask of pixels belonging to categorical class $c$. Let $\hat{y}_c : \Omega \to [0, 1]$ be a continuous predicted segmentation of a segmentation network that predicts the binary mask $y_c$ from an image. Connected component analysis (CCA) is used to extract individual instances of categorical class $c$ from binary masks and predicted segmentations. Each connected component in $y_c$ and $\hat{y}_c$ are identified with $\mathcal{I}_{y_c,n}$ and $\mathcal{I}_{\hat{y}_c,m}$, respectively, where $n$ and $m$ are the indices of the components.

In the proposed **Instance-wise loss**, segmentation quality is assessed for each ground truth instance by comparing it to the predicted segmentation instances that intersect with that ground truth instance. The Instance-wise loss ($\mathcal{L}_{instance}$) for class $c$ in a mini-batch of size $B$ is formalized in Equation (3), where $N$ is the total number of connected components in a ground truth image (i.e. total number of ground truth instances in the image) and $Z$ is the total number of connected components in all ground truth images in the mini-batch (i.e. total number of ground truth instances in the mini-batch).

$$\mathcal{L}_{instance} = \frac{1}{Z} \sum_{b=1}^{B} \sum_{n=1}^{N} \mathcal{L}_{seg} \left( \{\mathcal{I}_{\hat{y}_c,m} | \{\mathcal{I}_{\hat{y}_c,m} \cap \mathcal{I}_{y_c,n}\} \neq \emptyset\}_{m=1}^{M} , \mathcal{I}_{y_c,n} \right) \tag{3}$$

The proposed **Center-of-Instance** loss measures the segmentation quality of normalized instances, where the size and shape of each instance are normalized into a square (2D) or cube (3D) based on the center-of-mass, denoted as $\mathcal{C}(\mathcal{I}, \delta)$. The normalized size of the center-of-mass is controlled by the parameter $\delta$, where the default value is 1. For example, for $\delta = 3$, the size of center-of-mass will be $3 \times 3$ pixels in 2D or $3 \times 3 \times 3$ voxels in 3D. The Center-of-Instance loss ($\mathcal{L}_{center}$) is formalized in Equation (4). Note that $\mathcal{L}_{seg}$ in Equations (3) and (4) can be any segmentation losses (e.g. Dice loss (Milletari et al., 2016), Focal loss (Lin et al., 2017)).

$$\mathcal{L}_{\text{CIS}}(\delta) = \mathcal{L}_{seg} \left( \mathcal{C} \left( \{\mathcal{I}_{\hat{y}_c,m}\}_{m=1}^{M} , \delta \right) , \mathcal{C} \left( \{\mathcal{I}_{y_c,n}\}_{n=1}^{N} , \delta \right) \right) \tag{4}$$

## Appendix B. Formalism of Instance-wise Loss without True Positive Intersections with the Other Label Instances

This version of instance-wise segmentation loss still calculates segmentation quality individually for each instance in the ground truth image (similar to the original Instance-wise loss described above), but it does not include true positive intersections with other instances from the ground truth image. True positive intersections with other ground truth instances occur when one (relatively large) predicted segmentation instance intersects/segments multiple ground truth instances. This version of instance-wise segmentation loss can be formalized as Equation (5). Figure 4 shows visualization of both $\mathcal{L}_{instance}$ and $\mathcal{L}_{instance}^{-\text{TP}}$.

$$\mathcal{L}_{instance}^{-\text{TP}} = \frac{1}{Z} \sum_{b=1}^{B} \sum_{n=1}^{N} \mathcal{L}_{seg} \left( \{\mathcal{I}_{\hat{y}_c,m} | \{\mathcal{I}_{\hat{y}_c,m} \cap \mathcal{I}_{y_c,n}\} \neq \emptyset\}_{m=1}^{M} \setminus \{\mathcal{I}_{y_c,k} | k \neq n\}_{k=1}^{N} , \mathcal{I}_{y_c,n} \right) \tag{5}$$

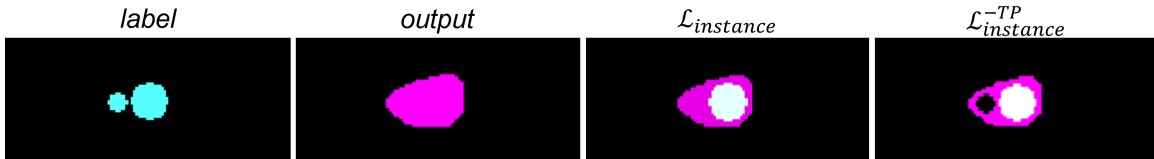

Figure 4: Comparison between $\mathcal{L}_{instance}$ from Equation (3) and $\mathcal{L}_{instance}^{-\text{TP}}$ from Equation (5).

Based on Equation (5) and visualization in Figure 4, the $\mathcal{L}_{instance}^{-\text{TP}}$ makes sure the predicted segmentation areas that intersect with other ground truth instances are not calculated as false positives. Based on our preliminary experiments, $\mathcal{L}_{instance}^{-\text{TP}}$ made the difference between the default weights (i.e., $a = 1, b = 1, c = 1$) and the best weights (i.e., $a = 1/4, b = 1/2, c = 1/4$) even smaller on the whole-image segmentation ($-1.1\%$ instead of $-2.3\%$ in DSC). Whereas, there were no significance differences in DSC between $\mathcal{L}_{instance}$ and $\mathcal{L}_{instance}^{-\text{TP}}$ when the best weights (i.e., $a = 1/4, b = 1/2, c = 1/4$) were used. This version of instance-wise segmentation loss can be used in our implementation of ICI loss by changing a single hyperparameter.

## Appendix C. Formalism of Dual Instance-wise loss

Our proposed ICI loss can be extended further into Dual Instance-wise and Center-of-Instance (DICI) loss by also calculating instance-wise segmentation loss for each predicted segmentation instance. The DICI loss can be formalized into Equation (6) which combines the global/pixel-wise segmentation loss ($\mathcal{L}_{global}$), Instance-wise loss for ground truth instance ($\mathcal{L}_{groundtruth}$), Instance-wise loss for predicted segmentation instance ($\mathcal{L}_{predicted}$), and Center-of-Instance loss ($\mathcal{L}_{center}$) with weights $a$, $b$, $c$, and $d$, respectively. The $\mathcal{L}_{predicted}$ can be calculated similarly to the $\mathcal{L}_{groundtruth}$ by using Equation (3) and changing the notations of $n$ and $N$ (for ground truth instances) to $m$ and $M$ (for predicted segmentation instances), and vice versa.

$$\mathcal{L} = a \times \mathcal{L}_{global} + b \times \mathcal{L}_{groundtruth} + c \times \mathcal{L}_{predicted} + d \times \mathcal{L}_{center} \qquad (6)$$

Based on our preliminary experiments, the DICI loss outperformed the ICI loss ($+1\%$ to $+2\%$ in DSC) when $\mathcal{L}_{groundtruth} = \mathcal{L}_{instance}^{-\text{TP}}$ with weights of $a = 1/4, b = 1/4, c = 1/4, d = 1/4$. However, note that the DICI uses more GPUs' memory and computation time than the ICI especially in the early epochs where there are many (false) predicted segmentation instances. The early implementation of DICI loss is available together with the implementation of ICI loss.

## Appendix D. 3D Residual U-Net from MONAI library package

```
from monai.networks.nets import UNet

model = UNet(
    spatial_dims=3,
    in_channels=1,
```

```
    out_channels=1,
    channels=(16,32,64,128,256),
    strides=(2, 2, 2, 2),
    num_res_units=2,
    norm=Norm.BATCH,
)
```

## Appendix E. Pseudo-code for Instance-wise loss ($\mathcal{L}_{instance}$)

---
**Algorithm 1:** Instance-wise loss ($\mathcal{L}_{instance}$)

---
**Input:** Extracted $cc\_label = x_1, \ldots, x_n$ and $cc\_output = y_1, \ldots, y_m$ from CCA
**Output:** $bwl$ /* the Instance-wise segmentation loss ($\mathcal{L}_{instance}$)      */
$bwl \leftarrow 0$
**for** $i \leftarrow 1$ **to** $n$ **do**
    $lbl_{x_i} \leftarrow$ label image with only $x_i$ blob /* other blobs are masked out      */
    $\mathbf{z} = [z_1, \ldots, z_p] \leftarrow$ blobs in $cc\_output$ that intersect with $x_i$
    **if** $\mathbf{z}$ *is not an empty list* **then**
        $out_{\mathbf{z}} \leftarrow$ output image with blobs in $\mathbf{z}$ /* other blobs are masked out      */
    **else**
        $out_{\mathbf{z}} \leftarrow$ output image with no blobs /* all blobs are masked out      */
    **end**
    $bwl \leftarrow bwl + \mathcal{L}_{seg}(out_{\mathbf{z}}, lbl_{x_i})$
**end**
$bwl \leftarrow bwl/n$

---

## Appendix F. Pseudo-code for Center-of-Instance loss ($\mathcal{L}_{center}$)

---
**Algorithm 2:** Center-of-instance segmentation loss ($\mathcal{L}_{center}$)

---
**Input:** *label* image, *output* image, extracted $cc_{label} = x_1, \ldots, x_n$ from *label* and
       $cc_{output} = y_1, \ldots, y_m$ from *output* by using CCA, and the new size $e$ for creating
       square/cube blobs
**Output:** $cbl$ /* the Center-of-Instance segmentation loss ($\mathcal{L}_{center}$)      */
$label_{cbl} \leftarrow label * 0$ /* copy label image with all zeros      */
$output_{cbl} \leftarrow output * 0$ /* copy output image with all zeros      */
**for** $i \leftarrow 1$ **to** $n$ **do**
    $com \leftarrow$ index of the center of the mass for $x_i$ blob
    $com_e \leftarrow$ indices of the newly transformed $com$ using the input parameter $e$
    $label_{cbl}(com_e) \leftarrow 1 + \epsilon$ /* $\epsilon$ is a smoothing parameter      */
**end**
**for** $i \leftarrow 1$ **to** $m$ **do**
    $com \leftarrow$ index of the center of the mass for $y_i$ blob
    $com_e \leftarrow$ indices of the newly transformed $com$ using the input parameter $e$
    $output_{cbl}(com_e) \leftarrow 1 + \epsilon$ /* $\epsilon$ is a smoothing parameter      */
**end**
$cbl \leftarrow \mathcal{L}_{seg}(output_{cbl}(com_e), label_{cbl}(com_e))$

---

## Appendix G. Computation Time

There are several sections in the implementation where additional computational times are needed for computing the proposed ICI loss. These sections are:

- **Connected component analysis (CCA)**: Based on kornia library's implementation, this function performs the max-pooling function with a fixed numbers of iteration. In our experiments for this study, we used 400 iterations for whole image experiments and 250 iterations for the patch-based experiments. For a 3D image with size of $192 \times 192 \times 192$, it takes roughly $0.4 - 0.6$ seconds to finish the CCA.

- **Iterations for accessing all instances in the label (ground truth) image for computing $\mathcal{L}_{instance}$**: Computational time needed for this section depends on the number of label instances in the label image. For accessing each label instance, it takes roughly $0.002 - 0.004$ seconds to finish.

- **Iterations for accessing all instances in the output (predicted segmentation) for computing $\mathcal{L}_{center}$**: Computational time needed for this section depends on the number of output instances in the output image (predicted segmentation). Note that early epochs might have more (false) predicted segmentation instances. For accessing each output instance, it takes roughly $0.002 - 0.004$ seconds to finish.

## Appendix H. Visualization of ICI loss in 3D

Figure 5 visualizes the transformation of instances of stroke lesions into cubes in 3D space for both the label and the prediction. In this visualization, one can appreciate how a small prediction (white blob in the (C) figure) is transformed into a cube of size of $7 \times 7 \times 7$ voxels (white cube in the (D) figure) like all other instances in the image.

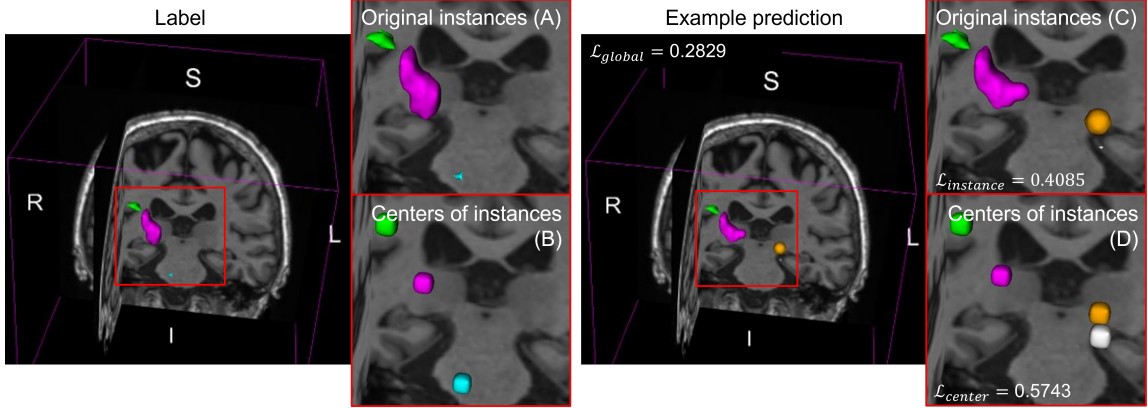

Figure 5: Visualization of ICI loss in 3D with $\mathcal{L}_{global}$ (0.2829), $\mathcal{L}_{instance}$ (0.4085), and $\mathcal{L}_{center}$ (0.5743) values produced by using Dice loss function Equation (2) are shown.

## Appendix I. Results of parameter search for deciding the optimum size of Center-of-Instance ($\mathcal{L}_{center}$)

Table 5: Validation measurements produced by 3D U-Net models trained for 200 epochs using data in full resolution (i.e., $192 \times 192 \times 192$) and the sigmoid function for segmentation (with a threshold of 0.5) with different sizes of Center-of-Instance.

| Weights | Size of Center | Mean Rank ($\downarrow$) | DSC ($\uparrow$) | Missed Instances ($\downarrow$) | False Instances ($\downarrow$) | Losses ($\downarrow$) Center | Blob | Best at Epoch |
|---|---|---|---|---|---|---|---|---|
| a=1; b=1; c=1 | 3 | 3.6 | 0.3625 [3] | 4.2 [3] | 9.6 [3] | 0.9261 [7] | 0.7725 [2] | 80 |
| a=1; b=1; c=1 | 5 | 4.0 | 0.3466 [5] | 4.9 [5] | **7.4 [1]** | 0.9203 [6] | 0.7921 [3] | 57 |
| a=1; b=1; c=1 | 7 | **2.2** | **0.3884 [1]** | 1.9 [2] | 8.7 [2] | 0.8841 [5] | **0.7532 [1]** | 41 |
| a=1; b=1; c=1 | 11 | 4.8 | 0.3437 [6] | 4.8 [4] | 75.0 [5] | 0.8550 [4] | 0.8016 [5] | 76 |
| a=1; b=1; c=1 | 15 | 3.2 | 0.3689 [2] | **1.8 [1]** | 16.6 [6] | 0.8487 [3] | 0.7936 [4] | 50 |
| a=1; b=1; c=1 | 31 | 4.2 | 0.3514 [4] | 4.9 [5] | 13.6 [4] | 0.7253 [2] | 0.8258 [6] | 84 |
| a=1; b=1; c=1 | 63 | 5.2 | 0.3292 [7] | 4.8 [4] | 21.5 [7] | **0.6285 [1]** | 0.8417 [7] | 95 |

## Appendix J. Other segmentation losses combined with the ICI loss

Table 6: Performance measurements produced by 3D U-Net models trained for 200 epochs using data in full resolution (i.e., $192 \times 192 \times 192$) and the sigmoid function for segmentation (with a threshold of 0.5) with different segmentation losses (other than Dice loss).

| Loss | Weights | Mean Rank | DSC | Mean MI | Mean FI |
|---|---|---|---|---|---|
| BCE Loss | a=1; b=0;c=0 | 2.3 | 0.2965 [3] | 2.0357 [3] | **2.8452 [1]** |
| + ICI Loss | a=1; b=1; c=1 | **1.3** | **0.3114 [1]** | **1.2083 [1]** | 6.7500 [2] |
| + ICI Loss | a=1/4; b=1/2; c=1/4 | 2.3 | 0.3027 [2] | 1.3571 [2] | 14.3869 [3] |
| Focal Loss | a=1; b=0;c=0 | 2.0 | 0.3115 [3] | 1.7619 [2] | **11.6607 [1]** |
| + ICI Loss | a=1; b=1; c=1 | 2.7 | 0.3215 [2] | 1.8571 [3] | 18.3869 [3] |
| + ICI Loss | a=1/4; b=1/2; c=1/4 | **1.3** | **0.3316 [1]** | **1.7440 [1]** | 13.0179 [2] |
| DiceCE Loss | a=1; b=0;c=0 | 2.7 | 0.3259 [3] | 1.2381 [3] | 3.8631 [2] |
| + ICI Loss | a=1; b=1; c=1 | **1.7** | 0.3356 [2] | 1.2083 [2] | **2.3214 [1]** |
| + ICI Loss | a=1/4; b=1/2; c=1/4 | **1.7** | **0.3452 [1]** | **0.9226 [1]** | 4.2024 [3] |
| DiceFocal Loss | a=1; b=0;c=0 | 2.3 | 0.3557 [3] | **1.2619 [1]** | 5.0119 [3] |
| + ICI Loss | a=1; b=1; c=1 | **1.7** | **0.3723 [1]** | 1.5833 [2] | 4.4345 [2] |
| + ICI Loss | a=1/4; b=1/2; c=1/4 | 2.0 | 0.3654 [2] | 1.9405 [3] | **2.4702 [1]** |
| GeneralizedDiceFocal Loss | a=1; b=0;c=0 | 3.0 | 0.3512 [3] | 1.5833 [3] | 6.3036 [3] |
| + ICI Loss | a=1; b=1; c=1 | 1.7 | 0.3686 [2] | 1.5119 [2] | **4.1190 [1]** |
| + ICI Loss | a=1/4; b=1/2; c=1/4 | **1.3** | **0.3933 [1]** | **1.4226 [1]** | 5.4405 [2] |

