# OpenReview forum: "Improving Segmentation of Objects with Varying Sizes in Biomedical Images using Instance-wise and Center-of-Instance Segmentation Loss Function"
_MIDL.io/2023/Conference — MIDL 2023 Oral_

### Official Review · Reviewer_AzAw · 2023-02-04

**Confidence:** 4
**Preliminary Rating:** 4
**Recommendation:** Poster

**Summary:**

This paper proposes a loss function that is designed to deal better with the segmentation of objects that have varying sizes. For that, different instances in the image contribute differently to the final loss, resulting in an instance-wise term, that is coupled with a global dice loss and third term that transforms every connected component into a small cube around its centroid (center-of-instance loss). Experiments are on binary segmentation of stroke lesions from MRI data from the ATLAS 2.0 challenge.

**Strengths:**

- Visual illustration of the idea in Figure 1 helps the reader to quickly understand the way in which the blob loss fails, and how this paper proposes to fix it.
- Experiments are on a public challenge, and results appear to be competitive with the test set leaderboard.

**Weaknesses:**

- The main concern I have is with the approach itself to the problem. This paper tries to improve the way we handle multiple sizes of objects in an image, which to me is quite clearly an **Instance Segmentation**-related problem, like when trying to segment all individual Multiple Sclerosis lesions in a brain scan, or all cells in a histopathological slide. In this case, the adequate metrics are very different* than the standard segmentation scenario, where one can normally use Dice or something similar. However, the authors have chosen to use a dataset of MRI with stroke lesions (the atlas challenge) in which, quoting the original paper**, "61.9% of subjects had only a single lesion". How is this data suitable to evaluate a loss function that is designed to allow us to handle instances of different sizes in an image?

^* https://arxiv.org/abs/2206.01653 ^**https://www.nature.com/articles/s41597-022-01401-7

- Another problem I find with this approach to image segmentation is that as soon as we are dealing with 3-dimensional data, we are all going to use patch-based training, with a patch resolution of, say, around 100x100x100. Indeed, the authors achievce much higher performance when they use patch-based training (table 4, DSC=0.58) than with whole image training (Table 2, DSC=0.44). In such small regions (patches), it does not seem like we are going to end up with multiple instances using the Atlas dataset, or that the variation in size of different instances, if we are "lucky" to have them in the same patch, is going to be noticeable. May the authors discuss this point?



**Deanonymize Review:**

no

**Detailed Comments:**

- Could you please increase font-size on Tables, the numbers are tiny and we probably do not need so many decimals.

- Shouldn't the authors also comment on the performance of the Atlas participants last Summer, which can be found here (https://atlas.grand-challenge.org/evaluation/lesion-segmentation-hidden-test-set/leaderboard/)? Their best result, 0.58 dice score, does not seem to be far away from the performance of the winner, and it would have ranked third in this competition, if I am reading correctly, which is positive.



**Paper Type:**

methodological development

**Questions To Address In The Rebuttal:**

Mainly, I would be pleased to hear the authors thoughts on why not to treat this as an instance segmentation problem and test their loss on instance segmentation problems, which seems much more natural. That would also lead to using different evaluation metrics than what was used in the Atlas challenge, where mostly there was a single lesion per scan.

Probably the best problem to try out their loss function is on surgical videos, where things change of size all the time, like segmenting polyps, or instrumets. I could suggest:
- Surgical instrument segmentation, see for example:
 Roß, T., Reinke, A., Full, P. M., Wagner, M., Kenngott, H., Apitz, M., ... & Maier-Hein, L. (2021). Comparative validation of multi-instance instrument segmentation in endoscopy: results of the ROBUST-MIS 2019 challenge. Medical image analysis, 70, 101920.

- Other suitable benchmarks could be related to nuclei instance segmentation on histologies, see for example:
CryoNuSeg: A dataset for nuclei instance segmentation of cryosectioned H&E-stained histological images, Amirreza Mahbod, Gerald Schaefer, Benjamin Bancher, Christine Löw, Georg Dorffner, Rupert Ecker, Isabella Ellinger; or the PanNuke dataset: https://jgamper.github.io/PanNukeDataset/

---

### Official Review · Reviewer_47Mc · 2023-02-04

**Confidence:** 4
**Preliminary Rating:** 5
**Recommendation:** Poster

**Summary:**

This paper develops the ICI loss (Instance wise centre of Instance) function to address the problem of learning in the context of instance segmentation. This loss is inspired from the blob loss and the lesion-wise loss and a weighted combination of an overall dice loss, a instance wise dice loss and a centre loss.
The ICI loss is compared with blob loss on the ATLAS  stroke segmentation challenge  and the different tested weighting are compared on both validation and test sets.

**Strengths:**

The paper is strongly motivated highlighting the limitations from the two other losses it gets inspiration from.
The explanation of the loss components is very well described with suitable depiction for an easier understanding
The experiments are well run and appropriately compared.

**Weaknesses:**

The choice of the optimal radius for the cube/square used for the optimal centre wise part of the loss is unclear and would require either an ablation study or some more justification - how would that relate to the  distribution of the objects of interest
All the experiments are performed in 3D and there is no indication of the stability of the weighting/ radius in 2D.


**Deanonymize Review:**

no

**Paper Type:**

methodological development

**Questions To Address In The Rebuttal:**

Please comment a bit further on the influence of square/cube radius and if known on optimal weighting in 2D.
Are there particular patterns of performance modification according to the weighting? It is a bit difficult from the tables alone to understand the influence of each part of the loss on the final result. The massive difference observed for the optimal choice in terms of missed instances lacks a bit of discussion

---

### Official Review · Reviewer_t4g1 · 2023-02-06

**Confidence:** 5
**Preliminary Rating:** 4
**Recommendation:** Oral

**Summary:**

This paper proposes a loss for semantic segmentation, that supervise each connected component separately. This is relevant in tasks such as brain lesions, where missing whole connected component has a clear impact on the clinical interpretation. It can also be an issue when components have a different size, where the small ones _could_ be ignored.

This works builds on [1], but refines it to better handle false positives in the background region. One main difference is that blob-loss relies only on pre-computed connected components, while the proposed losses compute CC on-the-fly for the predicted segmentation (it is not required to be derivable).

The results are good but not entirely convincing (hence the weak accept), despite the thorough evaluation and ablation studies. But I might have overlooked some key element, so I am open to be shown the right way during the rebuttal. At the same time, this paper really got me thinking, so it could easily upgrade my rating as this type of paper is valuable to the community.

---
[1] Kofler, Florian, et al. "blob loss: instance imbalance aware loss functions for semantic segmentation." arXiv preprint arXiv:2205.08209 (2022).

**Strengths:**

- Good explanation and illustration of key differences between the different works (I commend Fig 1)
- Clear experimental protocol, and reporting both validation curves (which, despite carying critical information, are too often missing) and test results
- Authors intend to share their code once the paper is de-anonymized

**Weaknesses:**

- Seemingly underwhelming results, compared to the initial claims. While the DSC and other metrics are improved, the number of missed instances remains roughly the same, which I think was one key motivation. I might have missed something/misinterpreted the reported metrics, see comments in "questions to address in the rebuttal"
- The paper could benefit from clearer formalism (see detailed comments)
- I suspect that a lot of the empirical findings are only valid with Dice loss as a base loss, and that alternative losses (most notably focal loss) could reach similar results without necessarily computing each connected component.

**Deanonymize Review:**

no

**Detailed Comments:**

### On formalism and loss definitions
I think the explanations could benefit from a bit more formalism, especially when writing the losses. Below is my own attempt, when rewriting both the blob-loss and the proposed ICI loss. Please correct me if I mis-understood any of those two, as the rest of my review rests on it.

Let us define $\Omega$ an image space (2D, 3D, this is not important here), $y: \Omega \rightarrow \\{0, 1\\}$ a ground truth for image $x$ and $s_\theta: \Omega \rightarrow [0, 1]$ the associated network prediction.

Let's also define $\Omega_y := \\{ i \in \Omega | y(i) = 1 \\} = \cup \left\\{\Omega_{y, n}\right\\}_{n=1}^N$ the subset of $\Omega$ where $y=1$.

$\Omega_{y,n}$ denotes each connected component $n$ of $y$. (It follows that they do not overlap.) $\Omega_{\theta,m}$ can be similarly defined for the $M$ connected components of the predicted segmentation.

At last, I will define a base-loss in the form $\mathcal L_\mathrm B(s_\theta, y ; S \subseteq \Omega) := \frac{1}{Z} \sum_{i \in S} s_\theta(i)y(i)$, which we denote sums over a subset $S$ and not the whole of $\Omega$.

Now, if I understood correctly, blob-loss can be written as follow:

$\\mathcal L_\\text{blob} :=\\sum_{n=1}^N \\mathcal L_\\mathrm B\left(s_\\theta, y ; \\Omega \\setminus (\\cup \\{ \\Omega_{y,n'}\\}_{n' \\neq n})\right)$ , i.e. for each connected component of $y$, summing over all $\Omega$ *except* the other connected components of $y$.

As correctly pointed out by the authors, the false positives of $s_\theta$ appear in each sub-computation of $\mathcal L_\mathrm B$. This goes even further:
$\\Omega \\setminus (\\cup \\{ \\Omega_{y,n'}\\}_{n' \\neq n}) = (\\Omega \\setminus \\Omega_y) \\cup \\Omega _{y,n}$,
which highlight perhaps more that the background is supervised $N$ times in total, which _could_ imbalance the training. (I also suspect that this makes the global dice loss used redundant.) This is the main limitation that the authors address in their contribution.

In contrast, the instance wise loss could be formalized as:

$\mathcal L_\text{IW} := \sum_{n=1}^N \mathcal L_\mathrm B \left(s_\theta, y; \Omega_{y,n} \cup \\{\Omega_{\theta,m} | \Omega_{y,n} \cap \Omega_{\theta,m} \neq \emptyset \\}_{m=1}^M \right) $

Here, only the "bare minimum" of pixels are included in each connected component supervision, and a global loss is required to supervise the background pixels without any false positive in it. The downside is that while all $\Omega_{y,n}$ can be pre-computed, $\Omega_{\theta,m}$ cannot which adds (expensive?) computations at training time.

Please correct me if I mis-represented any of those two losses.
(I am perfectly aware that I went overboard with all this, but this will also simplify some of my questions.)

---
Misc:
- I felt the "limitation" described in Section 2 for Inverse Weighting (IW) to be a bit "weak", and it could benefit from a bit more elaboration. Why assigning the weights once to each pixel (based on the ground truth instance sizes) not be enough? If I haven't mistaken things, blob-loss ends up supervising the background $N$ times ($N$ being the number of ground truth instances), which is, in a way, a re-weighting of those pixels.
- In Algorithm 1, I do not think that the $\verb|if else|$ ($z$ not being an empty set) is truly required?
- `A. Reinke 2021` is missing conference/journal/pre-print information. I think they have published several similar works at that time, so it would be best to clarify explicitly which one you are referring to

**Paper Type:**

methodological development

**Questions To Address In The Rebuttal:**

### Misc
- Can you provide a comparison of training iteration time for each method? As your proposed methods requires to compute "live" the connected components of the predicted segmentation
- I am wondering why the missed instance (MI) metrics is not reported on the testing set (Tables 2 and 4). Why is that?
- Moreover, in the end on the validation set, the Total MI is often worse than the blob loss (except for $a=b=c=1$). As supervising instances remains the main motivation of this work, does it not contradict your main claim? (Though yeah the lesion-wise f1 score is improved. Perhaps the paper could benefit from clarifying which metrics are the most clinically relevant ; I would believe that missing whole instances is worse that having a slightly worse segmentation.)
- Question for the blob-loss (but you are the experts here :-) ), but couldn't $\alpha$ be set to $0$, as the background is already supervised multiple times? (See detailed comments on why I think that.)
- I am personally not fond of Dice loss (for a whole variety of reasons), but here I will focus on a single one: computing it on a subset of the image is a bit of a grey-area, semantically speaking (we can compute it, but it does not always make complete sense). Other standard losses, such as cross-entropy or even focal-losses, are very straighforward when applied only to a subset of the image. My question is then (finally): would your (empirical) findings be any different, were another base loss be chosen over dice loss? (Cross-entropy, focal loss.)

---
### Center-of-instance loss
I remain a bit surprised on how the center of gravity for each instances is supervised.

Basically, the center of gravity for each instances $n$ and $m$ are computed (I assume with something like $\mathfrak C_{\theta,m} := \frac{\sum_{i \in \Omega_{\theta,m}}u{(i)}s_\theta{(i)}}{\sum_{i \in \Omega_{\theta,m}}s_\theta{(i)}}$, with $\Omega_{\theta,m} \subseteq \Omega$ the subset of pixels contained in component $m$ of $s_\theta$, $u{(i)}$ representing the 2-D coordinates of pixel $i$ and $s_\theta{(i)}$ the softmax predictions at pixel $i$. A similar computation can be done for ground truth $y$ at instance $n$.) Then a square of width $w$ is put on a synthetic map at location $\mathfrak C_{\theta,m}$, before computing a dice loss (or other) between the two "squared" maps. By doing so, the $w$ can have a big influence on the overall results (even more so with dice loss, which is sensitive when dealing with small objects), and can make things difficult to handle for multiple small and close instances. There is almost such a case in Appendix B, Figure 4, with the orange and white dots almost touching.

Why not simply minimize something like $\mathcal L_\text{CoI} := \frac{1}{N}\sum_n \min_m ||\mathfrak C_{\theta,m} - \mathfrak C_{y, n}||_2^2$, which I believe is minimized when $n=m$ and each center of gravity is at the "right" location.

---
**Final rating:** I have to say that I did a lot of pondering over this paper, which I believe is not finished yet. At the same time, it did foster a lot of interesting discussions during the rebuttal, which is one of the main goal of scientific conferences. Henceforth, I maintain my `weak accept` and confidence of `5`.

I believe that this paper would be a good candidate for a journal extension, with elements already discussed at many places during this rebuttal.

Cheers!

---

### Official Review · Reviewer_9ftw · 2023-02-06

**Confidence:** 4
**Preliminary Rating:** 4
**Recommendation:** Poster

**Summary:**

The paper presents a novel two-component loss for biomedical image segmentation tasks. They call it instance-wise and center-of-instance (ICI) and they claim it can be better than using the Dice loss.
They compare the ICI loss with the Dice loss and the blob loss in the task of stroke segmentation using a challenge from MICCAI 2022.
They report having improved performance of 1.7 to 3.7% compared to Dice on both validation and testing set by reporting the dice similarity coefficient.
The ICI is derived from a combination of the blob loss and the LesLoss losses.

**Strengths:**

The new loss presented in the paper is similar to the blob loss but with improvements. The computational time required is as well similar and the authors claim that it might be less dependent on parameter choice since the "simple choice" of parameters $a=b=c=1$ is already outperforming blob and dice losses.
Moreover the code will be available in Pytorch, which is a nice add-on for the community.

**Weaknesses:**

The authors mention the weaknesses already pretty nicely on the discussion. It would be nice to have seen how the loss performs with different sets of weights and not only staying with the simple choice. The computational time required for this new loss is still higher than other conventional losses.
The loss was only validated on one challenge.
It would have been nice to compare the loss to the state of the art lesLoss, despite the fact of the difficulty in performing this comparison.

**Deanonymize Review:**

no

**Paper Type:**

methodological development

**Questions To Address In The Rebuttal:**

1. How does the performance of the loss change with different sets of weights?
2. Could the loss be validated on multiple challenges or just one challenge?
3. Why wasn't the new loss compared to the state-of-the-art lesLoss, to Generalized Dice Loss, Cross Entropy Loss or Tversky Loss?

---

### Meta-Review · Area_Chair_57cp · 2023-02-24

**Recommendation:** Accept (Poster)
**Confidence:** 5

**Metareview:**

The paper presents a novel two-component loss for biomedical image segmentation, called instance-wise and center-of-instance (ICI) loss. The goal of the ICI loss is to improves the detection of small instances in image that contain large and small instances, depending on the image. Experiments include a comparison to the Dice loss and the blob loss in the task of stroke segmentation (challenge from MICCAI 2022).

Overall, reviews acknowledged the fact that:
- the paper is very clearly written and strongly motivated
- the experimental setup is sound and include a significant amount of results.
- the code will be shared.

Yet, several questions were also raised, dealing with the influence of hyperparameters, suggestions of formulation, clearing misunderstanding etc, to which the authors reply thoroughly with additional experiments and precise point-to-point response.

Hence I recommend acceptance.

Note that the questions led to several back and forth discussions with the reviewers (roughly 10000 words, around twice the original paper, the most extensive discussion I’ve seen!). Let me acknowledge the great commitment of MIDL reviewers at this point, because I believe they contributed a lot to improving this paper, not only in terms of its form, but also from a scientific perspective.
Nice to see that the scientific discussion has started even before the conference takes place!